# Observation of heat pumping effect by radiative shuttling

Yuxuan Li[1], Yongdi Dang[1], Sen Zhang[1], Xinran Li[1], Tianle Chen[1], Pankaj K. Choudhury ®[1], Yi Jin[1], Jianbin Xu ®[2], Philippe Ben-Abdallah ®[3] ✉, Bing-Feng Ju[4] & Yungui Ma ®[1] ✉

Heat shuttling phenomenon is characterized by the presence of a non-zero heat flow between two bodies without net thermal bias on average. It was initially predicted in the context of nonlinear heat conduction within atomic lattices coupled to two time-oscillating thermostats. Recent theoretical works revealed an analog of this effect for heat exchanges mediated by thermal photons between two solids having a temperature dependent emissivity. In this paper, we present the experimental proof of this effect using systems made with composite materials based on phase change materials. By periodically modulating the temperature of one of two solids we report that the system akin to heat pumping with a controllable heat flow direction. Additionally, we demonstrate the effectiveness of a simultaneous modulation of two temperatures to control both the strength and direction of heat shuttling by exploiting the phase delay between these temperatures. These results show that this effect is promising for an active thermal management of solid-state technology, to cool down solids, to insulate them from their background or to amplify heat exchanges.

Manipulating heat flows within a system is of prime importance for the development of a wide variety of technologies (microelectronics, energy conversion, building thermal control, satellite management, etc.). The nonlinearities of physical properties of materials with respect to the temperature can be taken advantage of for this purpose[1–3]. This nonlinear behavior has been exploited to manipulate heat flux in a similar way as currents in electrical circuits, enabling information processing, active thermal management, and even wireless sensing using heat as a primary source of energy with active thermal blocks such as thermal transistors[4–11], thermal diodes[10,12–18], thermal memories[19–24] and thermal logic gates[25,26]. These elements are the building blocks of a technology, also called "thermotronics" in analogy with traditional electronics, which allows a direct interaction of smart systems with their environment using thermal signals without external electricity supplying.

Many strategies have been proposed to date to actively control the heat flux and pump heat within a system and to develop smart sensors by exploiting external stimuli[27–36]. Also, a slow cycling modulation of control parameters or external fields near-topological singularities[37], such as exceptional points, have been used to enhance or inhibit energy exchange within a system as well as the geometric phase in non-reciprocal systems[38]. The spatiotemporal modulation of thermal properties, such as thermal conductivity, can also be used to control heat flux by giving rise to an effective convective component inside the system[39]. Finally, by periodically time-varying the temperature of two thermal baths connected to a system, the direction of heat flux flowing through it can also be controlled. In particular, when no thermal bias is present on average through the system, a thermal heat flux can cross it[40–42]. This effect is the so-called heat shuttling. The necessary condition for this phenomenon to occur

[1]The National Key Laboratory of Extreme Optics Technology and Instruments, Centre for Optical and Electromagnetic Research, College of Optical Science and Engineering; International Research Center (Haining) for Advanced Photonics, Zhejiang University, Hangzhou 310058, China. [2]Department of Electronic Engineering, The Chinese University of Hong Kong, Shatin, Hong Kong, China. [3]Laboratoire Charles Fabry, UMR 8501, Institut d'Optique, CNRS, Université Paris-Saclay, 2 Avenue Augustin Fresnel, 91127 Palaiseau, Cedex, France. [4]The State Key Lab of Fluid Power Transmission and Control, School of Mechanical Engineering, Zhejiang University, Hangzhou 310027, China. ✉e-mail: pba@institutoptique.fr; yungui@zju.edu.cn

is the presence of a nonlinear behavior within the system, which induces a symmetry breaking in the transport mechanism. This effect results from the local curvature of flux with respect to the temperature. When this curvature is negative, the system displays a negative differential thermal conductance[43], and the time modulation of the temperature tends to pump heat from cold to hotter parts of the system.

Recently, a radiative shuttling effect was predicted between two bodies made with materials having dielectric properties strongly temperature dependent such as phase change materials (PCMs)[28] or semiconductors[44–48]. But to date, no experimental proof of this effect has been reported. In this work, we present the experimental evidence. By probing the radiative heat flux exchanged in far-field regime between two parallel slabs based on a metal-insulator transition material coupled to temporally oscillating thermostats we show that the direction of average net heat flux the slabs exchange can be efficiently controlled by this time variation around the critical temperature of PCM. When the system has a negative differential thermal resistance, we show that the radiative shuttling can be used to insulate the two slabs from each other even in the presence of a temperature gradient, demonstrating that the shuttling effect acts in these conditions as a heat-pumping mechanism. We also explore the role of a simultaneous modulation of two thermostat temperatures on the control of both strength and direction of heat flux inside the system by leveraging the role of phase delay between the two thermostats.

## Results

To start let us consider the systems as sketched in Fig. 1 made with two parallel finite slabs based on PCMs which are separated by a gap $d = 0.5$ mm thick (this thickness is much larger than the thermal wavelength of slabs) of partial vacuum ($P \sim 10^{-4}$ Pa) and a view factor $F \sim 0.91$. In Fig. 1a, the left (L) slab is made of a n-doped silicon (Si) film of thickness $t = 200$ μm and of surface area $A = 10 \times 10$ mm$^2$ coated by magnetron sputtering with a vanadium dioxide (VO$_2$) thin film of thickness $e = 300$ nm, while the right (R) slab is a Si bulk sample coated by a black paint of emissivity $\varepsilon \sim 0.98$. In the second system, as sketched in Fig. 1b, the left slab is a multilayer Al/Si/VO$_2$ coated by a ZnS layer. In both systems, the temperature of right slab is hold constant at $T_0$ with a thermoelectric device and a Peltier element, while the temperature of left slab is modulated sinusoidally at a pulsation $\omega$ by Joule heating through the Si layer, which has an electrical resistivity $\rho \sim 0.01$ Ω cm around $T_0$ so that

$$T_L(t) = T_0 + \Delta T \sin(\omega t), \quad T_R(t) = T_0. \tag{1}$$

Both temperatures are monitored with thermistors, and the net radiative flux exchanged between the slabs is measured with a sensor (HS-10, Captec company) embedded inside the right slab. According to the radiometric theory, this flux reads

$$Q(t) = F\sigma\varepsilon(T_L)(T_L^4 - T_R^4), \tag{2}$$

where $\varepsilon(T_L)\frac{\varepsilon_L(T_L)\varepsilon_R(T_0)}{1-\rho_L(T_L)\rho_R(T_0)}$ is the effective emissivity of two slabs, which is expressed in term of average emissivity $\varepsilon_{L,R}(T_{L,0}) = \sigma^{-1}T_{L,R}^{-4}\int_0^{+\infty}I_\lambda^0(T_{L,0})\varepsilon_{\lambda,LR}(T_{L,0})d\lambda$ and of average reflectivity $\rho_{L,R}(T_{L,0})$, $I_\lambda^0(T)$ being the radiative intensity of a blackbody at temperature T and wavelength λ, σ the Stefan-Boltzmann constant and $\varepsilon_{\lambda,LR}$ the spectral emissivity of slabs which can be directly measured by a Bruker Fourier Transform Infrared Spectrometer. As the spectral reflectivity is concerned, it is deduced from Kirchhoff's law with additional transmission measurements.

For a weak temperature variation (i.e., $\delta T = \Delta T \sin(\omega t) \ll T_0$), the radiative flux exchanged between the slabs can be written in term of the thermal conductance

$$Q(t) = G(T_L)\delta T, \tag{3}$$

where $G(T_L) = 4F\sigma T_0^3\varepsilon(T_L)$ is the thermal conductance of heat exchange between the left and right body at temperature $T_L$. Written in term of transport properties at the (constant) temperature of right slab the flux reads

$$Q(t) \approx [G(T_0) + \delta T\dot{G}(T_0)]\delta T, \tag{4}$$

where $\dot{G}\frac{dG}{dT} = 4F\sigma T_0^3\dot{\varepsilon}$.

It turns out that the time averaged flux $\langle Q \rangle = \tau^{-1}\int_0^\tau Q(t)dt$ over one oscillation period $\tau = \frac{2\pi}{\omega}$ reads

$$\langle Q \rangle \approx \frac{(\Delta T)^2}{2}\dot{G}(T_0) \approx 2F\sigma\dot{\varepsilon}(T_0)T_0^3(\Delta T)^2. \tag{5}$$

It follows from this expression that the direction of average heat flux depends on the sign of the differential thermal conductance $\dot{G}$ which itself is proportional to the differential emissivity $\dot{\varepsilon}$ of the left body. This direction depends on the nature of materials which compose the slabs. When $\dot{\varepsilon} < 0$, heat is pumped (i.e. $\langle Q \rangle < 0$) from the right slab and transferred to the left slab. This situation occurs for instance in the system as sketched in Fig. 1a when $T_0$ is close to the critical temperature of PCM (see Fig. 2). Reversely, when $\dot{\varepsilon} > 0$, as in the system

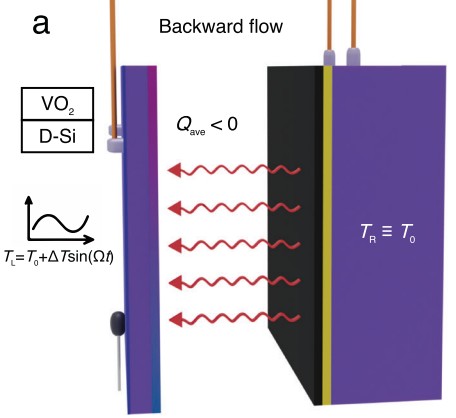

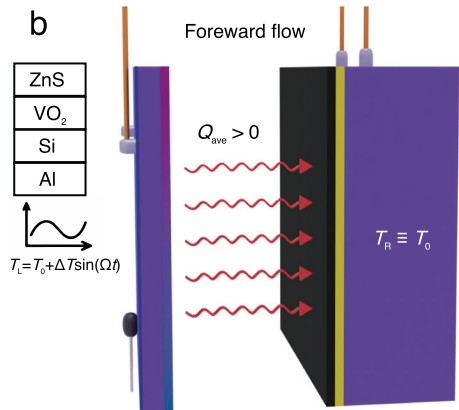

**Fig. 1 | Schematic of the heat shuttling model. a** The net heat flow is in the backward direction and **b** in the forward direction. The temperature of the left bath coated with VO$_2$ is periodically modulated while the right bath coated with the blackbody remains at a constant temperature $T_0$.

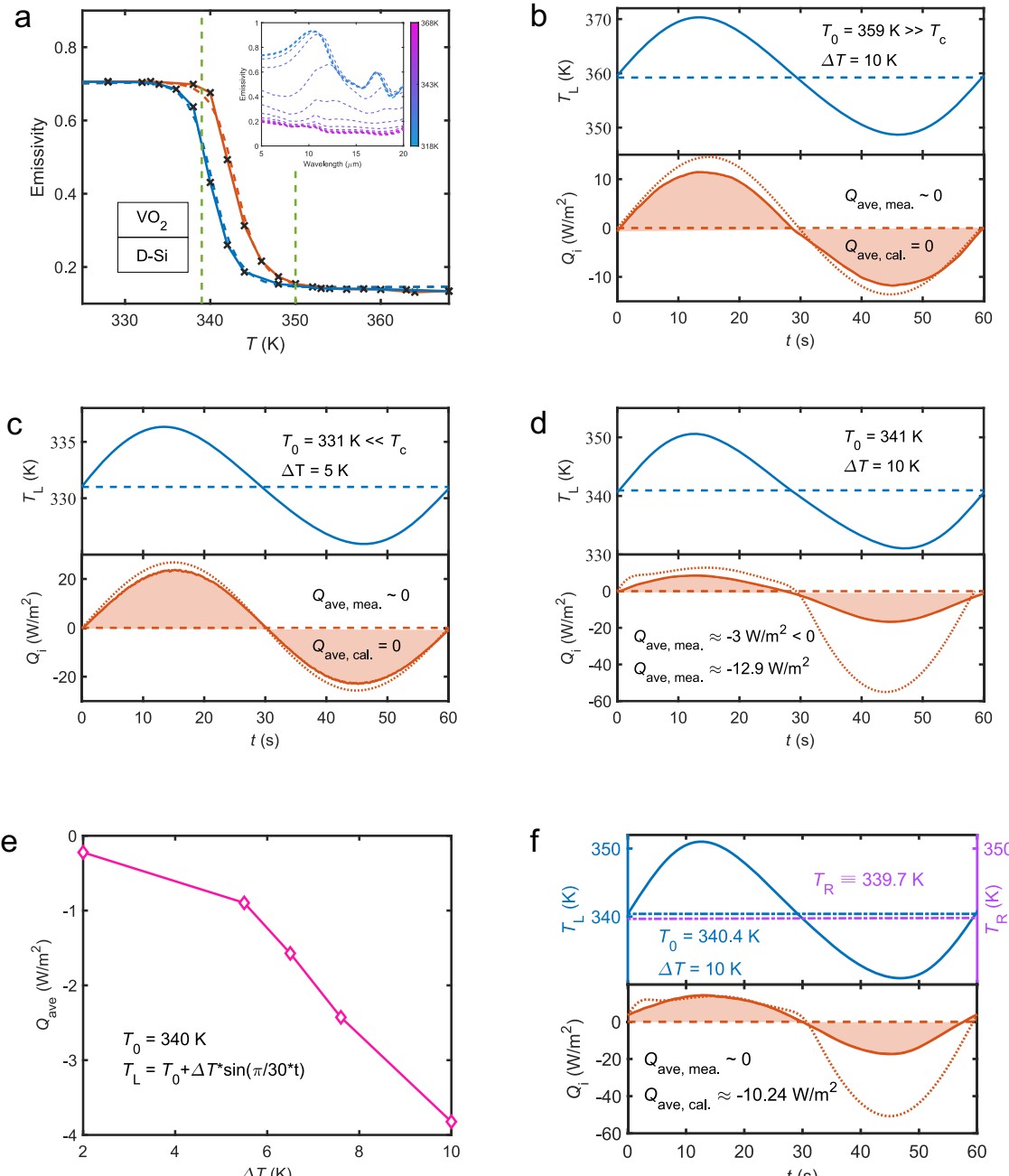

**Fig. 2 | Heat shuttling effect in the backward scenario. a** Emissivity of VO₂ (300 nm)/Si(200 μm) slab with respect to temperature during heating and cooling processes. Crosses correspond to experimental measurements. The inset gives the measured emissivity spectra at various temperatures. **b**–**d** Measured (solid line) and calculated (dotted line) temperature of the left slab and net heat flux between the two slabs during one oscillation period when $T_0 = 359$ K, 331 K, and 341 K, respectively. **e** Measured net average heat flux between the two slabs with respect to $\Delta T$ when $T_0 = 340$ K. **f** Thermal insulation by shuttling effect between the left slab of average temperature $T_0 = 340.4$ K and the right slab at fixed temperature $T_R = 339.7$ K when $\Delta T = 10$ K.

shown in Fig. 1b, $\langle Q \rangle > 0$ that is, the average net flux propagates from the hot to the cold slab (see Fig. 3).

It is worth noting that the average heat flux given by expression (5) is independent on the modulation frequency. This is implicitly related to the fact that this modulation takes place at a time scale, which is much larger than the thermalization time of left slab (i.e., adiabatic modulation). On the other hand, the magnitude of this flux depends quadratically on the amplitude $\Delta T$ of temperature oscillations and on the local slope of the emissivity with respect to the temperature. Hence, in a practical situation we can benefit from oscillating the temperature around the critical temperature of a phase-transition material, which is able to undergo an important change in its optical properties.

In the two systems investigated in this study, and shown in Fig. 1, the left slab is made of VO₂ films and the temperature modulation takes place around the critical temperature $T_c \sim 340$ K[49–51] of this material. In this region, the effective emissivity of slab drastically changes even with a tiny variation of the temperature. As shown in Fig. 2a (resp. Fig. 3a), we see that the emissivity contrast $\Delta \varepsilon$ is 0.55 (resp. 0.35) while the slope $|\dot{\varepsilon}_{max}|$ gets its maximal value at $T = 340$ K (resp. $T = 321$ K) for the cooling process and at $T = 343$ K (resp. $T = 326$ K) when the system is heated up.

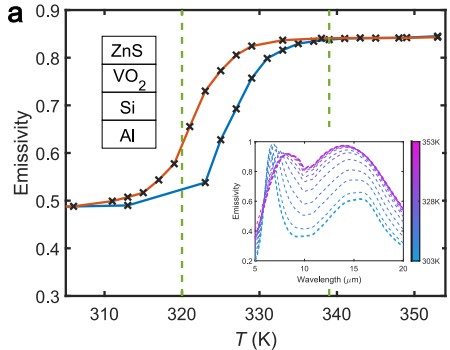
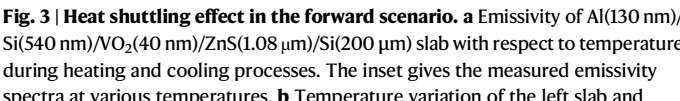
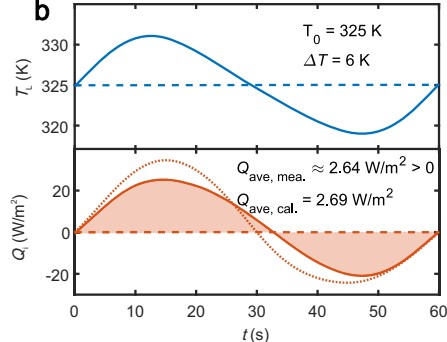

**Fig. 3 | Heat shuttling effect in the forward scenario. a** Emissivity of Al(130 nm)/Si(540 nm)/VO₂(40 nm)/ZnS(1.08 μm)/Si(200 μm) slab with respect to temperature during heating and cooling processes. The inset gives the measured emissivity spectra at various temperatures. **b** Temperature variation of the left slab and measured (solid line) and calculated (dotted line) neat heat flux exchanged between the two slabs during one period of oscillation when $T_0 = 325$ K and $\Delta T = 6$ K.

## Heat pumping and thermal insulation

Now, let us consider the system as sketched in Fig. 1a. In order to compare the measured heat flux exchanged between the slabs with the theoretical predictions, we first calculate the thermal emissivity of two slabs using the scattering matrix approach and Kirchhoff's law with the optical properties of material coming from the literature. Out of the transition region, we use the dielectric properties of VO₂ from Barker's measures[49] while the Looyenga mixing rule[52] is employed in the transition region, where the hysteresis response of VO₂ under periodical temperature modulation is modeled using the method described in ref. 53. For silicon, a Drude model is used to describe its dielectric permittivity with a plasma pulsation $\omega_p = 6.27 \times 10^{14}$ rad/s and collision (damping) frequency $\gamma = 1.15 \times 10^{13}$ rad/s[54]. The comparison between theory and measurements is summarized in Fig. 2 with the emissivity of the left slab (Fig. 2a) measured with a FTIR during both the heating and cooling processes. As shown in the inset of Fig. 2a the thermal emissivity is clearly a decaying function of the temperature. In Fig. 2b–d, we show the transient heat flux $Q$ measured for a temperature $T_L$ oscillating around different value of $T_0$ with a period of oscillation $\tau = 60$ s when data are collected at a frequency of about 3 Hz. When $T_0$ is distant from the transition region of PCM we see that the average net heat flux is almost equal to zero as predicted by expression (5). On the other hand, when $T_0$ is located in the transition region, the symmetry is broken in the system and a net heat flux is pumped on average from the right slab to the left slab. Hence, as shown in Fig. 2d when $T_0 = 341$ K and $\Delta T = 10$ K, the transient heat flows in the two half periods are clearly dissimilar from each other and lead to a nonzero average net heat flux ($<Q> \approx -3$ W/m²). In Fig. 2e, we check the influence of the oscillation amplitude $\Delta T$ around $T_0 = 340$ K. In agreement with expression (5) we see that the net heat flux increases monotically with $\Delta T$. Also, we demonstrate that the shuttling effect can be used to either pump heat or to simply insulate a solid from its background. Hence in Fig. 2f we see that, even in presence of a temperature gradient on average between the left (hot) slab and the right (cold) slab, i.e., $<T_L - T_R> = 0.7$ K, a thermal insulation can be induced by the shuttling effect. Notice that in the case where the average temperature $T_0$ is in the region where the dielectric properties of PCM bulk are significantly different than that of a film, an important discrepancy between the calculated value of the differential thermal emissivity $\dot\varepsilon(T_0)$ and its exact value can appear.

## Heat flux amplification

Reversely to the previous situation, the shuttling effect can also be used to amplify heat flux. This effect can be observed with an active slab highlighting a positive differential emissivity as with the structure shown in Fig. 1b and made of a multilayer 130 nm Al, 540 nm Si, 40 nm VO₂ and 1.08 μm ZnS films deposited on the same n-type silicon substrates as before[55]. The results of measurements and calculations are summarized in Fig. 3a. Unlike for the previous structure sketched in Fig. 1a, the measured average emissivity becomes this time an increasing function with respect to the temperature. Therefore, according to expression (5), the shuttling effect amplifies the transfer from the left slab to the right slab. As shown in Fig. 3b, when $T_0 = 325$ K and $\Delta T = 6$ K, a positive average shuttling flux $<Q> \approx 2.64$ W/m² has been measured. Similarly to the heat pumping, the amplification of heat flux can only be observed in the transition region of PCM (see Figs. 3a, b). The discrepancy observed in Figs. 2 and 3 between measurements and theoretical predictions can be attributed to the change of optical properties for the PCMs layer with respect to its thickness[56] and to the encapsulation of this layer. Notice that a negative or positive differential emissivity can also be achieved with VO₂-based metasurfaces[57].

## Shuttling induced by a simultaneous modulation of two reservoirs temperatures

Finally, we discuss the more general situation where the temperatures of two reservoirs are modulated periodically over time. To analyze this situation, we consider the case (see Fig. 4) where the left and right slabs are modulated with the same amplitude of modulation $\Delta T$ and frequency $\omega$ but with a phase delay $\Phi$. In this case the neat heat flux exchanged between the two slabs reads

$$Q \approx \left[ G(T_0) + G(T_0).\delta T \right](\delta T_L - \delta T_R), \qquad (6)$$

where $\delta T = (\delta T_L, \delta T_R)^t$ is the modulations vector with $\delta T_L = \Delta T \sin(\omega t)$ and $\delta T_R = \Delta T \sin(\omega t + \Phi)$.

It is straightforward to show that the averaged flux reads

$$\langle Q \rangle \approx \frac{(\Delta T)^2}{2}(1 - \cos \Phi)(\partial_L G - \partial_R G) \qquad (7)$$

with $\partial_L G \frac{\partial G}{\partial T_L} \frac{G(T_L, T_0) - G(T_0)}{\delta T_L}$ (resp. $\partial_R G \frac{\partial G}{\partial T_R} \frac{G(T_0, T_R) - G(T_0)}{\delta T_R}$). Notice that, by definition, $\partial_{L,R} G$ implicitly depends on the phase delay. For the same system as sketched in Fig. 1a, we see in Fig. 4c that this modulation leads to an average heat flux that is much larger than that with a single temperature oscillation (Fig. 2d). In particular, when the phase delay $\Phi = \pi$, we see that the measured average flux is about 10 times larger with $\Delta T = 10$ K. However, it is worthwhile to note that the direction of heat flux is independent on the phase delay. On the other hand, when the two slabs are identical and are PCM-based bilayers, the direction of heat flux can be controlled by an appropriate choice of $\Phi$. This

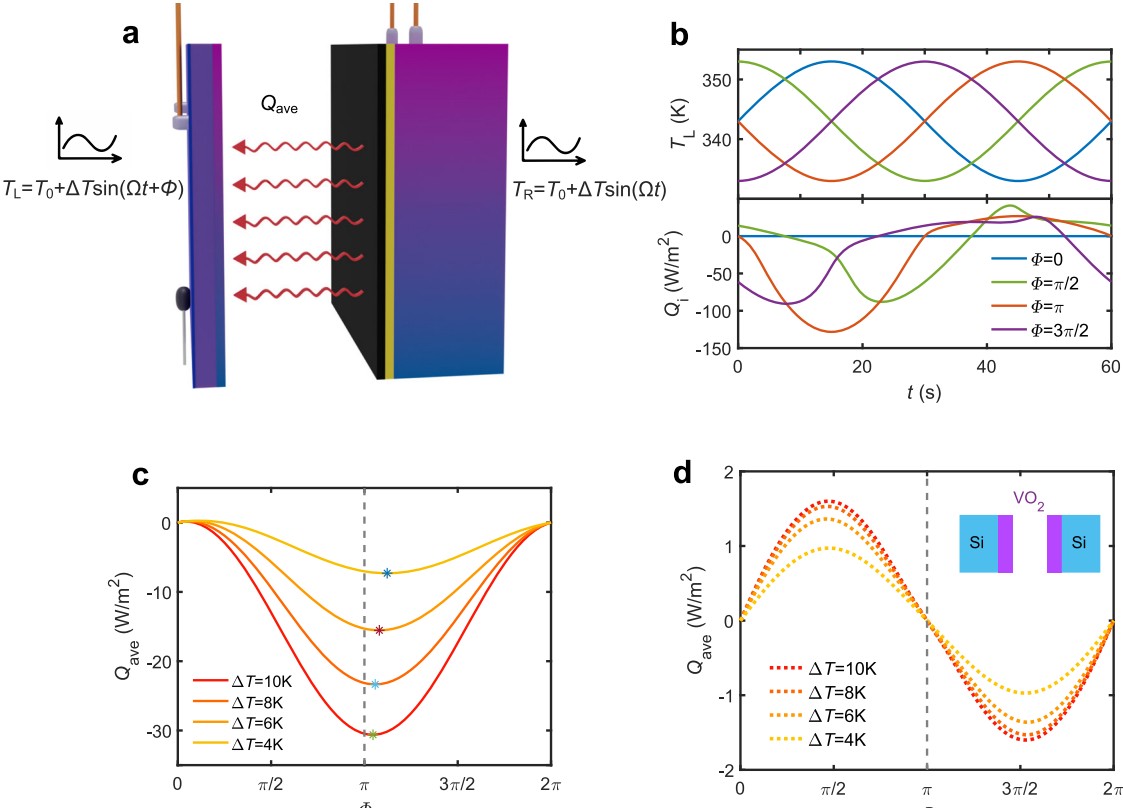

**Fig. 4 | Heat shuttling effect induced by a simultaneous modulation of temperatures of two slabs. a** Schematic of the mutual modulation. Both slabs are subject to a sinusoidal temperature modulation with the same frequency but with a phase delay. **b** Temporal evolution of the left slab temperature (top) and of net heat flux exchanged between the two slabs (low) similar to the system shown in Fig. 1a for different phase delay $\Phi$ when $T_0 = 343$ K and $\Delta T = 10$ K. **c** Average net heat flux with respect to $\Phi$ for different $\Delta T$. **d** Average net heat flux in a system made with two identical compounds VO$_2$(300 nm)/Si(200 μm).

situation is illustrated in Fig. 4d, for a system made with the compound VO$_2$ (300 nm)/ Si (200μm). In this case, the heat flux direction becomes switchable depending on the value of $\Phi$ and $|<Q>|$ reaches its maximum value at $\Phi \rightarrow \pi/2$ and $3\pi/2$. These results indicate that the phase delay can be used to tune both the amplitude and direction of the heat flux within symmetric system made with PCMs.

In conclusion, we have experimentally highlighted the radiative heat shuttling effect between two solids and demonstrated that this effect can be used to pump heat from the cold solid toward the hotter one, provided the latter displays a negative differential emissivity. We have shown that a prominent net heat flow can be generated by increasing the modulation amplitude of time-varying temperature in one solid, and we have demonstrated that the direction of heat flux can be tuned with the sign of the differential emissivity of the system. Finally, we have seen that the simultaneous modulation of temperatures of two reservoirs in contact with these solids brings an additional degree of freedom for controlling both the amplitude and the direction of average heat flux by tuning the phase delay between these two oscillations. This work paves the way for promising solutions in the field of active thermal management of solid-state systems. The radiative shuttling effect could be used to insulate two solids one from the other or to amplify the heat flux exchanged between a hot and a cold solid. The present work could be extended to the near-field regime where heat exchanges can be larger than the heat flux predicted by the Stefan Boltzmann's law (blackbody limit) by several orders of magnitude.

## Methods
### Evaluation of material parameters
For VO$_2$, before the phase change, the experimentally grown poly-crystalline VO$_2$ film is described by an isotropic effective

permittivity$\varepsilon_d = \frac{\varepsilon_\perp \pm \sqrt{\varepsilon_\perp^2 + 8\varepsilon_\perp \varepsilon_\parallel}}{4}$, where $\varepsilon_{\parallel,\perp} = \varepsilon_\infty + \sum_{j=1}^{N} \frac{S_j \omega_j^2}{\omega_j^2 - i\gamma_j\omega - \omega^2} \cdot \varepsilon_\parallel$ ($\varepsilon_\perp$) denotes the permittivity tensor parallel (perpendicular) to the (001)-axis of the tetragonal lattice of insulating VO$_2$, which is modeled as the sum of several Lorentz oscillators[49]. $\varepsilon_\infty$ is the permittivity at the infinite frequency; $S_j$, $\omega_j$ and $\gamma_j$ respectively denotes the oscillator strength, the phonon vibration frequency and the scattering rate. After the phase change, the metallic VO$_2$ is described by a Drude model: $\varepsilon_m = -\frac{\omega_p^2 \varepsilon_\infty}{\omega^2 + i\omega\gamma_p}$, where $\omega_p = 14000$ cm$^{-1}$ and $\gamma_p = 10000$cm$^{-1}$ are the plasma frequency and the scattering rate, respectively. For the permittivity of VO$_2$ films within the phase transition region, a simple Looyenga rule is used[53]: $\varepsilon_{eff} = (1-f)\varepsilon_d^{\frac{1}{3}} + f\varepsilon_m^{\frac{1}{3}}$ and $f(T) = \frac{1}{1 + \exp\left[\frac{W}{k_B}\left(\frac{1}{T} - \frac{1}{T_{half}}\right)\right]}$. $f$ is the temperature-dependent volume fraction of the metallic VO$_2$ domains within the film, where $W$ contains information about the width of temperature range of the phase transition region, and $T_{half}$ is the temperature at which half of the volume of the film is in the metallic state. For the calculation of the temperature dependence of the emissivity of the metasurface, $W = 3.79$ eV and $T_{half} = 339$ K are set corresponding to the suitable VO$_2$ film thickness and substrate.

### Sample fabrication
For the single layer VO$_2$ sample sketched in Fig. 1a, it was grown on a doped silicon substrate (10 × 10 × 0.2 mm$^3$) ultrasonically cleaned sequentially in acetone, methyl alcohol, and isopropyl alcohol. Each step was for 5 min. About 300-nm thick VO$_2$ film was deposited on the clean Si substrates with vanadium target by magnetron sputtering with

DC power of 200 W. During deposition, the chamber pressure was maintained at 5.5 mTorr with an Ar/O$_2$ mixed gas (70/4 sccm flow ratio). The sample was later heated to 450 °C for the formation of the VO$_2$ phase. For the multilayer sample sketched in Fig. 1b, the aluminum film (130 nm) was first magnetron sputtered on the doped silicon substrate at an Argon gas pressure 5.0 mTorr, and then the silicon film (540 nm) was deposited by electric beam evaporation. Later, the VO$_2$ film (40 nm) was deposited by the same technique as for the single layer sample. Lastly, the ZnS layer (1.08 μm) was deposited by electric beam evaporation.

## Heat flux measurement

The setup was placed inside a vacuum chamber with a gas pressure ~$10^{-4}$ Pa. The temperature of the right reservoir was maintained by a thermostat made of a thermoelectric device and a Peltier element. Blackbody paint (emissivity ~0.98) was coated on the thermal radiative exchange surface of the right reservoir. The two parts were separated at equal vacuum gaps of ~0.5 mm. The temperatures of two reservoirs were monitored by thermistors inserted into them. Heat flux lost or received by the blackbody with constant temperatures is measured using embedded sensors (HS-10, Captec Enterprise). The measurement sampling frequency for temperature and heat flux was ~3 Hz. The data were all recorded on a steady-state period response.

## Data availability

The data that support the findings of this study are available from the corresponding authors upon request.

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

## Acknowledgements

Y.G.M. thanks the partial support from the National Natural Science Foundation of China 62075196, Natural Science Foundation of Zhejiang Province LXZ22F050001 and DT23F050006, Leading Innovative and Entrepreneur Team Introduction Program of Zhejiang (2021R01001), and Fundamental Research Funds for the Central University (226-2024-00152). J.B.X. would like to thank RGC for support via AoE/P-701/20.

## Author contributions

Y.G.M. conceived and led the project. Y.X.L. conducted the simulation and experiment and wrote the draft. Y.D.D., S.Z., X.R.L., T.L.C., and Y.J. participated in the experiment and analyzed the data. P.K.C., J.B.X., and B.F.J. analyzed the data. P.B.A. participated in the initial theoretical discussions and the manuscript writing. All authors discussed the results and commented on the manuscript.

## Competing interests

The authors declare no competing interests.
