## [Peer Review File · Nature Communications]

Observation of Heat Pumping Effect by Radiative ShuttlingREVIEWER COMMENTS

Reviewer #1 (Remarks to the Author):

Please see the enclosed document.

Reviewer #2 (Remarks to the Author):

In this paper by Yuxuan Li et al, the authors reported on an experimental study of the heat shuttling phenomenon observed in a radiative system. Their study verifies the predictions of the radiative shuttling between two bodies having strong temperature-dependent dielectric properties. In addition, interesting heat pumping effects due to the radiative shuttling are demonstrated. The paper is well written, the study is timely, and the results are interesting. My general impression is that in this important field of thermal management, including thermal diodes, thermal transistors, absolute negative mobility etc. there are a lot of theoretical studies but few experimental demonstrations. So this work is welcome.

I have few considerations and questions which I would like the authors to address before publication.

- Line 61 and line 162: I would say that what it is actually studied in this paper is the effect of “the negative differential emissivity”, rather than “the negative differential thermal resistance”. Perhaps the connection between them should be clarified. In addition, the last sentence of the second paragraph (line 60-62) could be hard to understand. If this sentence refers to existing studies – which is what the reader expects since it is in the introductory paragraph – then the related literature should be cited.
(by the way, at line 58 probably the authors wanted to write ‘induces’ instead of ‘induced’).

-What effect is supposed to illustrate Fig. 2(f)? A heat pumping, the thermal insulation, or both? As “”, as indicated in Fig. 2(f), it seems that thermal insulation rather than heat pumping phenomenon is shown. If so it should be pointed out explicitly. In addition, in Fig. 2(f), it is also indicated that “”; why there is such a big difference between and ?

- Line 130: As it is indicated, for VO₂, why in Fig3. (a), the slope reaches its maximal value at , instead?

- Line 158: As indicated by Eq. (5), . How good is the agreement of experimental data with the theoretical prediction of Eq. (5)? What is the reason for deviations?

In addition, I have other minor suggestions:

a) Line 41: Ref. [44] should be cited here as a reference where the concept of the thermal transistor is put forward. Similarly, the paper by Terraneo et al [PRL 88, 094302, 2002], where the concept of thermal rectifier has been introduced for the first time, and the paper by B. Li et al [PRL 93, 184391 (2004)] should be cited.

b) Line 133: The information given here is for Fig. 2(a), but no information is provided for Fig. 3(a).

c) Line 161: The notation might be confusing. It has been defined in line 103. If here it refers

to , it should be indicated explicitly with a different notation.

d) Line 176: Figure 3 only contains two panels, (a) and (b); Is panel (c) missing?

e) Line 186: why Fig. 2(c) is cited here?

f) Line 440: To which system refers Fig. 4(d)?

My overall suggestion to the authors is to proofread the paper more carefully.

Giulio Casati

Review of the manuscript:

Observation of Heat Pumping Effect by Radiative Shuttling

This work deals with the experimental demonstration of the radiative shuttling effect previously reported in the literature. The obtained results are physically sound and could be used to pump in and pump out heat currents. Before the possible publication of their work, I recommend the auteurs to improve their manuscript by addressing the following remarks:

- **On VO₂ thermal hysteresis :** In deriving the model for the average heat flux in Eq. (5), the authors neglected to effect of the VO₂ thermal hysteresis. As this hysteresis is an intrinsic property of VO₂, the authors should consider its impact on the modeled and measured heat flux.
- **On the phase delay:** Authors experimentally showed that the phase delay of the temperature field can effectively be used to control de magnitude and direction of the average heat flux. The authors should support and describe this experimental evidence with a theoretical model.
- **On the modulation frequency:** which is the range of modulation frequencies for which the average heat flux is independent of the modulation frequency? Is the Stefan-Boltzmann equation valid for the considered temperatures varying with time?

Response Letter

The authors thank the two referees for their constructive remarks and suggestions. Here below are the detailed responses to their reports.

Reviewer #1 (Remarks to the Author):

This work deals with the experimental demonstration of the radiative shuttling effect previously reported in the literature. The obtained results are physically sound and could be used to pump in and pump out heat currents. Before the possible publication of their work, I recommend the auteurs to improve their manuscript by addressing the following remarks:

Comment 1: *On VO₂ thermal hysteresis: In deriving the model for the average heat flux in Eq. (5), the authors neglected to effect of the VO₂ thermal hysteresis. As this hysteresis is an intrinsic property of VO₂, the authors should consider its impact on the modeled and measured heat flux.*

Reply: Thanks for the question. The hysteresis behavior has been considered in our model taking into account the historical dependence of the VO₂'s emissivity on temperature. Ref. [53] (from our previous publications) has been cited to explain the details of the method. In lines 143-146, page 4, we have updated the original descriptions by the following words:

“...Out of the transition region, we use the dielectric properties of VO₂ from Barker's measures [49] while the Looyenga mixing rule [52] is employed in the transition region, where the hysteresis response of VO₂ under periodical temperature modulation is modeled using the method described in [53]...”.

Comment 2: *On the phase delay: Authors experimentally showed that the phase delay of the temperature field can effectively be used to control de magnitude and direction of the average heat flux. The authors should support and describe this experimental evidence with a theoretical model.*

Reply: We thank the referee for his/her suggestion. In lines 194-202, page 6, we have accordingly added the following sentences in the revised text:

In this case the neat heat flux exchanged between the two slabs reads

$$Q \approx [G(T_0) + \nabla G(T_0) \cdot \delta \mathbf{T}] (\delta T_L - \delta T_R), \quad (7)$$

where $\delta \mathbf{T} = (\delta T_L, \delta T_R)^t$ is the modulations vector with $\delta T_L = \Delta T \sin(\omega t)$ and $\delta T_R = \Delta T \sin(\omega t + \Phi)$.

It is straightforward to show that the averaged flux reads

$$\langle Q \rangle \approx \frac{(\Delta T)^2}{2} (1 - \cos \Phi) (\partial_L G - \partial_R G) \quad (8)$$

with $\partial_L G \equiv \frac{\partial G}{\partial T_L} \approx \frac{G(T_L, T_0) - G(T_0)}{\delta T_L}$ (resp. $\partial_R G \equiv \frac{\partial G}{\partial T_R} \approx \frac{G(T_0, T_R) - G(T_0)}{\delta T_R}$). Notice that, by definition, $\partial_{L,R} G$ implicately depends on the phase delay.

Comment 3: *On the modulation frequency: which is the range of modulation frequencies for which the average heat flux is independent of the modulation frequency? Is the Stefan-Boltzmann equation valid for the considered temperatures varying with time?*

Reply: The Stefan Boltzmann's law remains valid as long as the system is at local thermal equilibrium and the applicability conditions of fluctuation dissipation theorem for the fluctuating currents are respected. Practically speaking the temperature could be modulated at high frequency (modulations at gigahertz frequency have already been achieved with nanostructures). But if the temperatures of electrons and phonons are different, the adiabaticity condition which is the key assumption in the shuttling is not respected anymore and the theoretical framework used in the present work should be extended.

Reviewer #2 (Remarks to the Author):

In this paper by Yuxuan Li et al, the authors reported on an experimental study of the heat shuttling phenomenon observed in a radiative system. Their study verifies the predictions of the radiative shuttling between two bodies having strong temperature-dependent dielectric properties. In addition, interesting heat pumping effects due to the radiative shuttling are demonstrated. The paper is well written, the study is timely, and the results are interesting. My general impression is that in this important field of thermal management, including thermal diodes, thermal transistors, absolute negative mobility etc. there are a lot of theoretical studies but few experimental demonstrations. So this work is welcome.

Reply: We thank the referee for his/her positive assessment.

I have few considerations and questions which I would like the authors to address before publication.

Comment 1- *Line 61 and line 162: I would say that what it is actually studied in this paper is the effect of “the negative differential emissivity”, rather than “the negative differential thermal resistance”. Perhaps the connection between them should be clarified. In addition, the last sentence of the second paragraph (line 60-62) could be hard to understand. If this sentence refers to existing studies – which is what the reader expects since it is in the introductory paragraph – then the related literature should be cited.(by the way, at line 58 probably the authors wanted to write ‘induces’ instead of ‘induced’).*

Reply: The referee is right. The shuttling effect is related to the presence of a negative differential emissivity. We have shown that this concept is closely related to the presence of a negative thermal conductance in the system. To clarify this we have rephrased the paragraph written between lines 102 and 112, pages 3 and 4 in the revised manuscript as followed (we have also removed all references to negative differential resistances to make the above message clearer):

“... ”

For a weak temperature variation (i.e. $\delta T = \Delta T \sin(\omega t) \ll T_0$)

the radiative flux exchanged between the slabs can be written in term of the thermal conductance

$$Q(t) = G(T_L)\delta T, \quad (4)$$

where $G(T_L) = 4F\sigma T_0^3 \varepsilon(T_L)$ is the thermal conductance of heat exchange between the left and right body at temperature T_L . Written in term of transport properties at the (constant) temperature of right slab the flux reads

$$Q(t) \approx [G(T_0) + \delta T \dot{G}(T_0)]\delta T, \quad (5)$$

where $\dot{G} \equiv \frac{dG}{dT} = 4F\sigma T_0^3 \dot{\varepsilon}$.

It turns out that the time averaged flux $\langle Q \rangle = \tau^{-1} \int_0^\tau Q(t)dt$ over one oscillation period $\tau = \frac{2\pi}{\omega}$ reads

$$\langle Q \rangle \approx \frac{(\Delta T)^2}{2} \dot{G}(T_0) \approx 2F\sigma \dot{\varepsilon}(T_0) T_0^3 (\Delta T)^2. \quad (6)$$

...”

Comment 2- What effect is supposed to illustrate Fig. 2(f)? A heat pumping, the thermal insulation, or both? As s indicated in Fig. 2(f), it seems that thermal insulation rather than heat pumping phenomenon is shown. If so it should be pointed out explicitly. In addition, in Fig. 2(f), it is also indicated that why there is such a big difference between and ?

Reply: Fig. 2(f) shows the *insulation* effect induced by the shuttling when $T_L > T_R$. Since, in this case, T_0 is close to the lower edge of the transition region of PCM a tiny difference between the dielectric properties of bulk and that of a film can significantly modify the value of the differential thermal emissivity $\dot{\varepsilon}(T_0)$. In lines 167-170, page 5, of the revised manuscript we have added the following sentences to outline this point:

“... Notice that in the case where the average temperature T_0 is in the region where the dielectric properties of PCM bulk are significantly different than that of a film, an important discrepancy between the calculated value of the differential thermal emissivity $\dot{\varepsilon}(T_0)$ and its exact value can appear. ...”

In addition, the caption of Fig. 2(d) has replaced by the following: Thermal insulation by shuttling effect between the left slab of average temperature $T_0 = 340.4$ K and the right slab at fixed temperature $T_R = 339.7$ K when $\Delta T = 10$ K.

Comment 3- Line 130: As it is indicated, for VO₂, why in Fig3. (a), the slope reaches its maximal value at , instead?

Reply: This was caused by a mistake forgetting to separately describe the emissivity slope of the two samples. In lines 131-137, page 4, the original sentences have been rephrased by the following:

“...In the two systems investigated in this study, and shown in Fig.1, the left slab is made of vanadium dioxide (VO₂) films and the temperature modulation takes place around the critical

temperature $T_c \sim 340$ K [49-51] of this material. In this region, the effective emissivity of slab drastically changes even with a tiny variation of the temperature. As shown in Figs. 2(a) (resp. Fig.3(a)), we see that the emissivity contrast $\Delta\epsilon$ is 0.55 (resp. 0.35) while the slope $|\dot{\epsilon}_{\max}|$ gets its maximal value at $T = 340$ K (resp. $T = 321$ K) for the cooling process and at $T = 343$ K (resp. $T=326$ K) when the system is heated up. ...”

Comment 4- Line 158: As indicated by Eq. (5), how good is the agreement of experimental data with the theoretical prediction of Eq. (5)? What is the reason for deviations?

Reply: The main reason for the deviations we observed between experiments and theoretical predictions is mainly due to a dependence of the optical properties of PCM with the thickness of deposited layers and also to its dependence with the surrounding materials. In the absence of available data for the dielectric property of VO₂ we have used for our theoretical calculations the dielectric properties of bulk material. This has been specified in the revised manuscript.

Comment 5: In addition, I have other minor suggestions:

a) Line 41: Ref. [44] should be cited here as a reference where the concept of the thermal transistor is put forward. Similarly, the paper by Terraneo et al [PRL 88, 094302, 2002], where the concept of thermal rectifier has been introduced for the first time, and the paper by B. Li et al [PRL 93, 184391 (2004)] should be cited.

Reply: These pioneer works have been cited at the beginning of revised manuscript as the following:

1. N. Li, J. Ren, L. Wang, G. Zhang, P. Hänggi & B. Li. Phononics: Manipulating heat flow with electronic analogs and beyond. Rev. Mod. Phys. 84, 1045 (2012).
2. M. Terraneo, M. Peyrard & G. Casati. Controlling the Energy Flow in Nonlinear Lattices: A Model for a Thermal Rectifier. Phys. Rev. Lett. 88, 094302, (2002).
3. B.W. Li, L. Wang & G. Casati. Thermal Diode: Rectification of Heat Flux. Phys. Rev. Lett. 93, 184301 (2004).

b) Line 133: The information given here is for Fig. 2(a), but no information is provided for Fig. 3(a).

Reply: Information relations to Fig. 3(a) have been added

c) Line 161: The notation might be confusing. It has been defined in line 103. If here it refers to , it should be indicated explicitly with a different notation.

Reply: Noticed with thanks. The gradience for the average temperature between the feft and right objects has been redefined by the expression : $\langle T_L-T_R \rangle = 0.7$ K.

d) Line 176: Figure 3 only contains two panels, (a) and (b); Is panel (c) missing?

Reply: This typo has been corrected. Here we refer to Figs. 3(a) and 3(b).

e) Line 186: why Fig. 2(c) is cited here?

Reply: It is Fig.2(d). This typo has been corrected and the text has been clarified as well as the caption of Fig.4

f) Line 440: To which system refers Fig. 4(d)?

Reply: This has been specified in the figure caption. It corresponds to the net flux exchanged between two identical compounds VO₂(300 nm)/Si(200μm).

My overall suggestion to the authors is to proofread the paper more carefully.

Reply: The manuscript has been carefully proofread.

REVIEWERS' COMMENTS

Reviewer #1 (Remarks to the Author):

The authors have properly addressed my previous comments, so I recommend the publication of their work.

Reviewer #2 (Remarks to the Author):

The authors have modified the paper according to my recommendations and therefore I recommend acceptance for its publication. There is only a minor detail: refs. [2, 3] are listed only in the bibliography, but they are not referred in the text. They should be cited on line 41.

Responses to the referees' comments

To Reviewer #1 (Remarks to the Author):

Comment: The authors have properly addressed my previous comments, so I recommend the publication of their work.

Answer: Thanks so much!

To Reviewer #2 (Remarks to the Author):

Comment: The authors have modified the paper according to my recommendations and therefore I recommend acceptance for its publication. There is only a minor detail: refs. [2, 3] are listed only in the bibliography, but they are not referred in the text. They should be cited on line 41.

Answer: The error has been corrected in the revised version.